

**Interhemispheric Effect of Global Geography on Earth's**
**Climate Response to Orbital Forcing**
**Rajarshi Roychowdhury[1] and Robert DeConto[1]**
[1]University of Massachusetts - Amherst
*Correspondence to:* rroychowdhur@geo.umass.edu
**Postal Address:**
Department of Geosciences
627 North Pleasant Street
233 Morrill Science Center
University of Massachusetts
Amherst, MA 01003-9297
**Abstract**
The climate response of the Earth to orbital forcing shows a distinct hemispheric
asymmetry due to the unequal distribution of land in the Northern versus Southern
Hemispheres. This asymmetry is examined using a Global Climate Model (GCM) and a
Land Asymmetry Effect (LAE) is quantified for each hemisphere. The results show how
changes in obliquity and precession translate into variations in the calculated LAE. We
find that the global climate response to specific past orbits is likely unique and modified
by complex climate-ocean-cryosphere interactions that remain poorly known and difficult



to model. Nonetheless, these results provide a baseline for interpreting contemporaneous
proxy climate data spanning a broad range of latitudes, which maybe especially useful in
paleoclimate data-model comparisons, and individual time-continuous records exhibiting
orbital cyclicity.

## 1.    Introduction

The arrangement of continents on the Earth's surface plays a fundamental role in the
Earth's climate response to forcing. This global "geography" is primarily the result of the
horizontal and vertical displacements associated with plate tectonics. While these
processes are ongoing, the global continental configuration has been close to its present
form since the mid-Cenozoic. Today, more continental land area is found in the Northern
Hemisphere (68%) as compared to the Southern Hemisphere (32%). These different
ratios of land vs. ocean in each hemisphere affect the balance of incoming and outgoing
radiation, atmospheric circulation, ocean currents, and the availability of terrain suitable
for growing glaciers and ice-sheets. As a result of this land-ocean asymmetry, the
climatic responses of the Northern and Southern Hemisphere differ for an identical
change in radiative forcing (Barron et al., 1984; Deconto et al., 2008; Kang et al., 2014;
Loutre, 2003; Short et al., 1991).
A number of classic studies have shown interhemispheric asymmetry in climate response
of Northern and Southern Hemispheres. Climate simulations made with coupled
atmosphere-ocean GCMs typically show a strong asymmetric response to greenhouse-gas
loading, with Northern Hemisphere high latitudes experiencing increased warming
compared to Southern Hemisphere high latitudes (Stouffer et al., 1989; Flato and Boer,



2001). GCMs also show that the Northern and Southern Hemispheres respond differently
to changes in orbital forcing (e.g. Philander et al., 1996). While the magnitude of
insolation changes through each orbital cycle is identical for both hemispheres, the
difference in climatic response can be attributed to the fact that Northern Hemisphere is
land-dominated while Southern Hemisphere is water dominated (Croll, 1870). This
results in a stronger response to orbital forcing in the Northern Hemisphere relative to the
Southern Hemisphere.
The changing continental configurations as a result of plate tectonics have been linked
with climate change over a wide range of timescales (e.g. Crowley and North, 1996;
DeConto, 2009; Fawcett and Barron, 1998; Hay, 1996). The distribution of continents
and oceans have an important effect on the spatial heterogeneity of the Earth's energy
balance, primarily via the differences in albedos and thermal properties of land versus
ocean (Trenberth et al., 2009). The latitudinal distribution of land has a dominant effect
on zonally averaged net radiation balance due to its influence on planetary albedo and
ability to transfer energy to the atmosphere through long-wave radiation, and fluxes of
sensible and latent heat. The latitudinal net radiation gradient controls the total poleward
heat transport requirement, which is the ultimate driver of winds, and ocean circulation
(Stone, 1978).
Oceans have a relatively slower response to seasonal changes in insolation due to the
higher specific heat of water as compared to land, and mixing in the upper ~10-150 m of
the ocean. As a result, in the ocean-dominated Southern Hemisphere, the surface waters
suppress extreme temperature swings in the winter and provide the atmosphere with a



source of moisture and diabatic heating. In the land-dominated Northern Hemisphere, the
lower heat capacity of the land combined with relatively high albedo results in greater
seasonality, particularly in the interiors of large continents of Asia and North America.
The continentality of the Northern Hemisphere manifests itself in different
hemispherically asymmetric climatic phenomenon, like the well-known Asian monsoonal
circulation system. The intertropical convergence zone (ITCZ) is considered to be the
region of low-level convergence and convective precipitation. The ITCZ moves further
away form the equator during the Northern summer than the Southern one due to the
continentiality of the Northern Hemisphere (Kang et al., 2008; Philander et al., 1996).
The land surface available in a particular hemisphere also affects the potential for
widespread glaciation. The extreme cold winters associated with large continents provide
the means of accumulation of winter snow, while the critical factor for formation of ice-
sheets is annual ablation and can be estimated by the sum of Positive Degree Days (PDD)
in a year (e.g. Huybers, 2006).
Continental geography has a strong impact on polar climates, as is evident from the very
different climatic regimes of the Arctic and the Antarctic. Several early paleoclimate
modeling studies using GCMs investigated continental distribution as a forcing factor of
global climate (e.g. Barron et al., 1984; Hay et al., 1990). These studies demonstrated that
an Earth with its continents concentrated in the low latitudes is warmer and has lower
equator-to-pole temperature gradients than an Earth with only polar continents. Although
these early model simulations did not incorporate all the complexities of the climate
system, the results provided valuable insights from comparative studies of polar versus



equatorial continents in the Earth and showed that changes in continental configuration
has significant influence on climatic response to forcing.
**2.    Methods**
**2.1    Experimental design**
We use the latest (2012) version of the Global ENvironmental and Ecological Simulation
of Interactive Systems (GENESIS) 3.0 GCM with a slab ocean component (Thompson
and Pollard, 1997) rather than a full-depth dynamical ocean (Alder et al., 2011). The slab-
ocean predicts sea surface temperatures and ocean heat transport as a function of the local
temperature gradient and the zonal fraction of land versus sea at each latitude. While
explicit changes in ocean currents and the deep ocean are not represented, the
computational efficiency of the slab-ocean version of the GCM allows numerous
simulations with idealized global geographies and greatly simplifies interpretations of the
sensitivity tests by precluding complications associated with ocean model dependencies.
In addition to the atmosphere and slab-ocean, the GCM includes model components
representing vegetation, soil, snow, and thermo-dynamic sea ice. The 3-D atmospheric
component of the GCM uses an adapted version of the NCAR CCM3 solar and thermal
infrared radiation code (Kiehl et al., 1998) and is coupled to the surface components by a
land-surface-transfer scheme (LSX). In the setup used here, the model atmosphere has a
spectral resolution of T31 (~3.75°) with 18 vertical layers. Land-surface components are
discretized on a higher resolution 2°x2° grid.
The GCM uses various geographical boundary conditions (described below) in 2°x2° and
spectral T31 grids for surface and AGCM models, respectively. For each set of



experiments, the model is run for 50 years. Spin-up is taken into account, and equilibrium
is effectively reached after about 20 years of integration. The results used to calculate
interhemispheric effects are averaged over the last 20 years of each simulation.
Greenhouse gas mixing ratios are identical in all experiments and set at preindustrial
levels with $CO_2$ set at 280 ppmv, $N_2O$ at 288 ppbv and $CH_4$ at 800 ppbv. The default
values for $CFCl_3$ and $CF_2Cl_2$ values are set at 0 ppm. The solar constant is maintained at
1367 $Wm^{-2}$.
**2.2    Asymmetric and symmetric Earth geographies**
The GCM experiments are divided into three sets: 1) Preindustrial CONTROL 2)
NORTH-SYMM and 3) SOUTH-SYMM.  The Preindustrial CONTROL experiments use
a modern global geography spatially interpolated to the model's 2°x2° surface grid
(Koenig et al., 2012). The geography provides the land-ice sheet-ocean mask and land–
surface elevations used by the GCM.
To simulate the climate of an Earth with meriodionally symmetric geographies, we
created two sets of land surface boundary conditions: NORTH-SYMM and SOUTH-
SYMM.  For the NORTH-SYMM experiments, the CONTROL experiment boundary
conditions are used to generate a modified GCM surface mask, by reflecting the Northern
Hemisphere geography (land-sea-ice mask, topography, vegetation, soil texture) across
the equator into the Southern Hemisphere. Similarly, in the experiment SOUTH-SYMM,
the land mask and geographic boundary conditions in the Southern Hemisphere are
mirrored in the Northern Hemisphere.  The NORTH-SYMM and SOUTH-SYMM



boundary conditions are shown in Figure 1B and 1C, with the CONTROL (Fig. 1A) for
comparison.
**3.        Asymmetry in the Earth's climate**
We begin our study by investigating the asymmetry in the Earth's climate. In our first
experimental setup, we run the GCM with modern day orbital configuration, i.e.
eccentricity is set at 0.0167, obliquity is set at 23.5° and precession such that perihelion
coincides with Southern Hemisphere summer. Figure 2A shows the present day summer
insolation intensity and Figure 2B shows present day Summer Energy for reference. The
Summer Energy (J) is defined as defined as:
$J = \sum_i \beta_i (W_i \times 86,400)$                                    …(1)
where $W_i$ is mean insolation measured in W/m$^2$ on day i, and β equals 1 when Wi ≥ τ and
zero otherwise. τ = 275 W/m$^2$ is taken as the assumed threshold for melting of ice at the
Earth's surface). Mean Summer Temperatures (ST) are calculated from the GCM as the
mean of the average daily temperatures for the summer months in each hemisphere (JJA
in Northern Hemisphere; DJF in Southern Hemisphere). Figure 2C shows the mean
summer temperature for a simulation with modern orbit. The zonal averages (calculated
for each latitude) demonstrate the inherent asymmetry in the Earth's climate between
Northern and Southern Hemispheres, especially evident in the higher latitudes. A better
indicator of the Earth's climate system, which quantifies both the intensity of summer as
well as the duration of the melt season, is the sum of Positive Degree Days (PDD). The
sum of Positive Degree-Days is calculated as:





$PDD = \sum_i \propto_i T_i$ ...(2)
where $T_i$ is the mean daily temperature on day i, and α is one when $T_i \geq 0°C$ and zero
otherwise. The PDD captures the intensity as well as the duration of the melt season, and
has been shown to be indicative of the ice-sheet response to changes in external forcing.
Figure 2D shows the PDD for modern orbit, and the zonal averages are plotted in the log
scale. The extreme asymmetry between the Northern and Southern Hemispheres
observed in the summer temperatures is also evident in the calculated PDDs.
The observed asymmetry in the Northern and Southern Hemispheres can be attributed to
three primary causes: (i) variation in insolation intensity across the Northern and
Southern Hemispheres caused by the precession of the equinoxes (today perihelion
coincides with January 3, just after the December 21 solstice, leading to slightly stronger
summer insolation in the Southern Hemisphere); (ii) the effect of the continental
geography on climate; and (iii) the effect of interhemispheric continental geography on
climate, i.e. the effect of Northern Hemisphere continental geography on Southern
Hemisphere climate and vice-versa. Here, we attempt to isolate the effect of
interhemispheric continental geography on climate (i.e. cause (iii) above) by comparing
results from GCM simulations using modern versus idealized (hemispherically
symmetric) global geographies (Fig. 1).
Next, we maintain a modern orbit to test the effect of meriodionally symmetric continents
(Fig. 2E-H). Figure 2E and 2F show the summer temperature and PDD from a simulation
in which the Northern Hemisphere geography is reflected in the Southern Hemisphere
(thus making the Earth geographically symmetric). Figure 2G and Figure 2H shows the



summer temperature and PDD from a hypothetical simulation with symmetric Southern
Hemisphere continents. Symmetric continents make the climates of Northern and
Southern Hemispheres almost symmetric (>95%), with some small remaining asymmetry
due to the current timing of perihelion with respect to the summer solstices.
The simulations with modern and idealized (symmetric) geographies are used to quantify
the different climate responses to a range of orbits. By comparing the climatic response
from simulations with different geographies, we isolate and estimate the effect of
interhemispheric continental geography and the influence of one hemisphere's geography
on the climate response of the opposite hemisphere.
**3.1    Effect of Southern Hemisphere on Northern Hemisphere climate**
To estimate the effect of Southern Hemisphere continental geography on the Northern
Hemisphere, we compare the NH climate from the CONTROL simulation (asymmetric,
modern orbit) and NORTH-SYMM (symmetric Northern continents in both
hemispheres). In these simulations, the only difference in setup is the Southern
Hemispheric continental distribution. Thus the differences in Northern Hemisphere
climate from the two simulations, if any, can be safely ascribed as the 'effect of Southern
Hemisphere continental geography on Northern Hemisphere climate'. We quantify this
interhemispheric effect of Southern Hemisphere continental geography on NH climate as:
$$e_{\widehat{Summer\ Temp}} = \frac{1}{n}\sum_{i}^{n}(T_i^{control} - T_i^{north}) \qquad \dots(3)$$
$$\widehat{e_{PDD}} = PDD^{control} - PDD^{north} \qquad \dots(4)$$

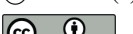



where $T_i^{control}$ and $PDD_i^{control}$ are the mean daily temperature on day i and PDD from the
control simulation, and $T_i^{North}$ and $PDD_i^{North}$ are the mean daily temperature on day i and
PDD from the simulation with the North-symmetric geography. 'n' is the number of days
in the summer months in each hemisphere (JJA in Northern Hemisphere; DJF in
Southern Hemisphere)
Figure 3A and 3B show the effect of Southern Hemisphere continental geography on
Northern Hemisphere summer temperature and PDD respectively. For the Northern
Hemisphere, the summer temperatures are calculated over the months of June, July, and
August when the insolation intensity over the Northern Hemisphere is strongest. The
asymmetry in the Southern Hemisphere landmasses leads to weakening of the summer
warming over North America and Eurasia (blue shaded regions correspond to cooling).
Consequently, summer temperatures over Northern Hemisphere continents are lower by
3-6°C relative to a symmetric Earth. There is a positive warming effect in the North-
Atlantic Ocean, and in general the Northern Hemisphere oceans are slightly warmer
relative to a symmetric Earth. The general trends in the interhemispheric effect on PDD
(Fig. 3B) mimic those of the summer temperatures (Fig. 3A).
**3.2      Effect of Northern Hemisphere on Southern Hemisphere climate**
Similarly, we estimate the effect of Northern Hemisphere continental geography on
Southern Hemisphere by comparing the SH climate of the CONTROL simulation
(asymmetric, modern orbit) and the SOUTH-SYMM (symmetric southern continents in
both hemispheres). In these simulations, the differences in Southern Hemisphere climate
in the CONTROL and SOUTH-SYMM simulations, if any, can be ascribed as the 'effect

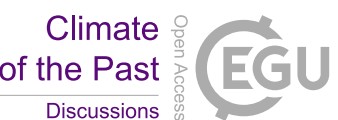

of Northern Hemisphere continental geography on Southern Hemisphere climate'. We
quantify this interhemispheric effect of Northern Hemisphere continental geography on
SH climate as:
$$e_{\widehat{Summer\,Temp}} = \frac{1}{n}\sum_i^n (T_i^{control} - T_i^{south}) \qquad \dots(5)$$
$$\widehat{e_{PDD}} = PDD^{control} - PDD^{south} \qquad \dots(6)$$
where $T_i^{control}$ and $PDD_i^{control}$ are the mean daily temperature on day i and PDD from the
control simulation, and $T_i^{south}$ and $PDD_i^{south}$ are the mean daily temperature on day i and
PDD from the simulation with the south-symmetric geography.
Figure 3C and 3D show the effect of Northern Hemisphere continental geography on
Southern Hemisphere summer temperature and PDD, respectively. For the Southern
Hemisphere, the summer temperatures are calculated over the months of December,
January, and February when the insolation is most intense during the year. Southern
Hemisphere landmasses, except Antarctica, generally show a cooling response during
summer, due to Northern Hemisphere geography. Over Antarctica, summer temperatures
are higher in the control simulations than in the symmetric simulations, leading to the
inference that there is a warming (increase) in summer temperatures due to
interhemispheric effect. Also, the Southern Ocean shows a strong positive temperature
effect (warming) relative to a symmetric Earth, although this Southern Ocean response
might be different or modified if a full-depth dynamical ocean model were used.





**4.**     **Interhemispheric effect on the Earth's climate response to orbital (astronomical)**
**forcing**
Next, we examine the effect of the opposite hemisphere on the Earth's climate response
to changes in obliquity (axial tilt) and precession (positions of the solstices and equinoxes
in relation to the eccentric orbit). The orbital parameters used in these experiments are
idealized and do not correspond to a specific time in Earth's history. Rather, they are
chosen to provide a useful framework for studying the Earth's climate response to
precession and obliquity. HIGH and LOW orbits approximate the highest and lowest
obliquity in the last three million years (Berger and Loutre, 1991).  NHSP (Northern
Hemisphere Summer at Perihelion) and SHSP (Southern Hemisphere Summer at
Perihelion) orbits correspond to Northern and austral summers coinciding with
perihelion, respectively, and represent the two extreme configurations of precession, with
obliquity set at its mean value averaged over the last 3 million years. Eccentricity is set at
the same moderate value (mean eccentricity over the last 3 million years) for all
simulations. Table 1 summarizes the orbits used in the ensemble of model simulations.
Here, we focus only on the sum of the Positive Degree Days (PDD) calculated from our
simulations. PDD is a better indicator of air temperature's influence on annual ablation
over ice-sheets than summer temperature, since this metric captures both the intensity and
duration of the melt season.
**4.1     Interhemispheric effect on precessional (cycle) response of the Earth's climate**
Changes in precession primarily affect seasonal insolation intensity that is well known to
be out-of-phase in both hemispheres (e.g. Raymo et al., 2006). The out-of-phase summer





energy (J) variation is shown in Figure 4A for reference. In one precessional cycle lasting
~23-kyr, the perihelion position of the Earth's orbit moves from the Northern
Hemisphere summer solstice (NHSP) to the Southern Hemisphere summer solstice
(SHSP), which are also the two extreme precessional configurations. We run the
simulations at these two extreme precessions, keeping all other orbital parameters
constant at their mean values. The difference in the calculated PDDs from the two
simulations (represented as $\Delta PDD_{precession}$) gives an estimate of the Earth's climate
response to the combined effect of the two precessional motions (wobbling of the axis of
rotation and the slow turning of the orbital ellipse). Figure 4B shows the precessional
response of the Earth in terms of PDD, and it is observed that the Northern and Southern
Hemisphere responses are not symmetrical. Running the same simulations with a North-
symmetric Earth (Fig. 4C) and a South-symmetric Earth (Fig. 4D) results in a nearly
symmetrical climate responses to the precessional cycle.
**4.2    Interhemispheric effect on obliquity (cycle) response of the Earth's climate**
In contrast to precession, obliquity alters the seasonality of insolation equally in both
hemispheres (Fig. 4E). A reduction in the tilt from 24.5° (HIGH) to 22° (LOW) reduces
annual insolation by ~17 W/m$^2$ and summer insolation by ~45 W/m$^2$ in the high latitudes.
In the tropics, summer insolation increases by up to ~5 W/m$^2$. Loutre et al. (2004) among
others predicted that global ice volume changes at the obliquity periods could be
interpreted as a response to mean annual insolation and meridional insolation gradients.
Similar to the experimental setup described above, we ran two simulations with the
highest and lowest axial tilts, keeping all other orbital parameters constant at their mean
values. The difference in the calculated PDDs (represented as $\Delta PDD_{obliquity}$) provides an



estimate of the Earth's climate response to changes in tilt. Figure 4F shows $\Delta PDD_{obliquity}$

and the zonal averages reveal the asymmetry in the climate response to obliquity.

Running the same simulations with a North-symmetric Earth (Fig. 4G) and a South-

symmetric Earth (Fig. 4H) produces a nearly symmetrical climate response to the

obliquity cycle.

**5.    Quantification of the Land Asymmetry Effect (LAE)**

**5.1    Effect of Southern Hemisphere geography on Northern Hemisphere climate**

The effect of Southern Hemisphere continental geography on Northern Hemisphere at the

two extreme precessional orbits is estimated using the same method described above,

with Interhemispheric effect of Southern Hemisphere continental geography on NH

climate at 'NHSP' calculated as:

$$(\widehat{e_{PDD}})_{NHSP} = PDD_{NHSP}^{control} - PDD_{NHSP}^{north} \qquad \qquad …(7)$$

and interhemispheric effect of Southern Hemisphere continental geography on NH

climate at 'SHSP' calculated as:

$$(\widehat{e_{PDD}})_{SHSP} = PDD_{SHSP}^{control} - PDD_{SHSP}^{north} \qquad \qquad …(8)$$

Figure 5A shows the spatial variation of $(\widehat{e_{PDD}})_{NHSP}$. The Northern Hemisphere

landmasses show a strong negative response to PDD when perihelion coincides with

Northern Hemisphere summer (NHSP). In this orbit, the Northern Hemisphere

experiences elevated summer insolation, but the response is weakened due to the

interhemispheric effect. This dampening effect is greatest in the interiors of the Northern



Hemisphere continents (Fig. 5A). According to Milankovitch theory, the Northern
Hemisphere should experience 'interglacial' conditions when perihelion coincides with
boreal summer. However, because of the interhemispheric effect, interglacial (warm
summer) conditions are muted relative to those on a symmetric Earth. Figure 5B shows
the spatial variation of $(\widehat{e_{PDD}})_{SHSP}$. When perihelion coincides with Southern
Hemisphere summer (SHSP), the Northern Hemisphere continents have a weak positive
effect, leading to slightly warmer conditions relative to a symmetric Earth.
Next we try to observe the interhemispheric effect on $\Delta PDD$ for a *transition* from SHSP to NHSP
orbit. Thus the Interhemispheric effect of Southern Hemisphere continental geography on
Northern Hemisphere response to a precession cycle is:
$(\widehat{e_{PDD}})_{precession} = \Delta PDD^{control}_{precession} - \Delta PDD^{north}_{precession}$                    …(9)
The calculated effect is plotted spatially in Figure 6A, and shows a strong negative effect
on Northern Hemisphere PDDs. For the Northern Hemisphere, the transition from SHSP
to NHSP equates to a transition from cool to warm climate. The negative
interhemispheric effect decreases the $\Delta PDD$ in the real Earth, thus weakening the effect
of precession on the Northern Hemisphere.
The effect of Southern Hemisphere continental geography on NH climate response at the
two extreme obliquity orbits are estimated as:
$(\widehat{e_{PDD}})_{HIGH} = PDD^{control}_{HIGH} - PDD^{north}_{HIGH}$                    …(10)
and





$(\widehat{e_{PDD}})_{LOW} = PDD_{LOW}^{control} - PDD_{LOW}^{north}$ …(11)
At HIGH obliquity, there exists a negative effect on Northern Hemisphere continents
(Fig. 5C), which mutes the strong insolation intensity during summer months. In the
Northern Hemisphere, as a result of continental asymmetry, a decrease in the equator to
pole temperature gradient is observed. A lowering of summer temperatures and
temperature gradient due to the interhemispheric effect has a negative impact on the
deglaciation trigger associated with HIGH obliquity orbits. Thus the interhemispheric
effect would hinder the melting of ice during high-obliquity orbits. At LOW obliquity,
the negative effect over Northern Hemisphere continents is generally less intense (Fig.
5D). However, even the modest lowering of summer temperatures caused by the
interhemispheric effect would support the growth of ice sheets during low obliquity
orbits.
Further, we calculate the interhemispheric effect on $\Delta PDD$ for a transition from LOW to
HIGH orbit (obliquity cycle). This Interhemispheric effect of Southern Hemisphere
continental geography on Northern Hemisphere response to an obliquity cycle is:
$(\widehat{e_{PDD}})_{obliquity} = \Delta PDD_{obliquity}^{control} - \Delta PDD_{obliquity}^{north}$ …(12)
The calculated effect is spatially plotted in Figure 6C, and shows a small negative effect
in the high latitudes, and a positive effect in the low latitudes. The transition from LOW
to HIGH corresponds to a transition from cold to warm climate. The negative
interhemispheric effect decreases the $\Delta PDD$, thus weakening the climate response of
obliquity cycle in the high latitudes. The positive interhemispheric effect increases the



$\Delta PDD$, thus strengthening the climate response of obliquity cycle in the Northern
Hemisphere low latitudes.
**5.2      Effect of Northern Hemisphere geography on Southern Hemisphere climate**
The effect of Northern Hemisphere continental geography on SH climate response at two
extreme precessional orbits is estimated as:
$(\widehat{e_{PDD}})_{NHSP} = PDD_{NHSP}^{control} - PDD_{NHSP}^{south}$                                              …(13)
and
$(\widehat{e_{PDD}})_{SHSP} = PDD_{SHSP}^{control} - PDD_{SHSP}^{south}$                                              …(14)
The spatial variation of $(\widehat{e_{PDD}})_{NHSP}$ is shown in Figure 5E. During NHSP orbit, the
Southern Hemisphere experiences 'glacial' (cold summer) conditions due to the weaker
summer insolation. The positive effect in the Southern Hemisphere leads to weaker
cooling relative to a symmetric Earth. Thus, when perihelion coincides with Northern
Hemisphere summer, the interhemispheric effect dampens the magnitude of 'glacial'
versus 'interglacial' conditions in both hemispheres. When perihelion coincides with
Southern Hemisphere summer (SHSP), the southern high latitudes experience intense
summer insolation.  The positive warming effect (Fig. 5F) amplifies the 'interglacial'
conditions in the Southern Hemisphere, predicted by Milankovitch theory.
The interhemispheric effect on $\Delta PDD$ for a transition from SHSP to NHSP orbit, or the
interhemispheric effect of Northern Hemisphere continental geography on Southern
Hemisphere response to a precession cycle is:



$$(\widehat{e_{PDD}})_{precession} = \Delta PDD_{precession}^{control} - \Delta PDD_{precession}^{south} \qquad \ldots(15)$$
The calculated effect is plotted spatially in Figure 6B, and shows a positive effect on
PDD over Southern Hemisphere high latitudes. For the Southern Hemisphere, the
transition from SHSP to NHSP equates to a transition from warmer to cooler climate. The
positive interhemispheric effect at high latitudes decreases the $|\Delta PDD|$ in the real Earth,
thus weakening the effect of precessional cycle in the Southern Hemisphere high
latitudes.
The interhemispheric effect of Northern Hemisphere continental geography on Southern
Hemisphere climate at the two extreme obliquity configurations is calculated as:
$$(\widehat{e_{PDD}})_{HIGH} = PDD_{HIGH}^{control} - PDD_{HIGH}^{south} \qquad \ldots(16)$$
and
$$(\widehat{e_{PDD}})_{LOW} = PDD_{LOW}^{control} - PDD_{LOW}^{south} \qquad \ldots(17)$$
The spatial variations of $(\widehat{e_{PDD}})_{HIGH}$ and $(\widehat{e_{PDD}})_{LOW}$ are shown in Figure 5G and 5H,
respectively. In the Southern Hemisphere, the positive interhemispheric effect on PDD
over Antarctica and the Southern Ocean leads to overall higher temperatures in the high
southern latitudes as compared to a symmetric Earth. During high obliquity orbits, this
positive effect contributes to deglaciation and during low obliquity orbits; the positive
effect (warming) hinders the growth of ice sheets.
Lastly, we calculate the interhemispheric effect on $\Delta PDD$ for a transition from LOW to
HIGH orbit (obliquity cycle):



$$(\widehat{e_{PDD}})_{obliquity} = \Delta PDD_{obliquity}^{control} - \Delta PDD_{obliquity}^{south} \qquad \ldots(18)$$
The calculated effect is plotted in Figure 6D, and shows largely a negative effect in the
Southern Hemisphere, with a positive effect in the high latitudes. The transition from
LOW to HIGH corresponds to a transition from cold to warm climate. The positive
interhemispheric effect increases the $\Delta PDD$, thus amplifying the effect of obliquity over
Antarctica.
**6.     Conclusions**
The unbalanced fraction of land in the Northern versus Southern Hemisphere has
remained almost unchanged for tens of millions of years. However, the significance of
this continental asymmetry on Earth's climate response to forcing has not been
previously quantified with a physically based climate models. We find that continental
geography has an important control on the climate system's response to insolation
forcing, and this may help explain the non-linear response of the Earth's climate to
insolation forcing.
According to classical Milankovitch theory, the growth of polar ice sheets at the onset of
glaciation requires cooler summers in the high latitudes, in order for snow to persist
throughout the year. During warm summers at the high latitudes, the winter snowpack
melts, inhibiting glaciation or leading to deglaciation if ice sheets already exist. Thus, the
intensity of summer insolation at high latitudes, especially the Northern polar latitudes,
has been considered the key driver of the glacial-interglacial cycles and other long-term
climatic variations. At precessional periods, at which the high latitude summer intensity



primarily varies, the land asymmetry effect plays an important role by amplifying (or
weakening) the effect of summer insolation intensity.
In all the orbital configurations simulated here, we find that the geography of the
Southern Hemisphere weakens the temperature response of the high Northern
Hemisphere latitudes to orbital forcing. Consequently, this leads to a larger latitudinal
gradient in summer temperatures in the Northern Hemisphere compared to that of a
symmetric Earth. In particular, the amplification (or weakening) of the response to
insolation changes at precessional and obliquity periods might explain some of the
important features of late Pliocene-early Pleistocene climate variability, when obliquity-
paced cyclicity dominated precession in global benthic $\delta^{18}$O records. In Figure 6, we have
demonstrated that the interhemispheric effect causes a suppression of the effects of
precessional cycle on the Earth's surface. In other words, the real Earth has a smaller
response to a precession cycle as compared to the hypothetical symmetric Earth. We have
also showed that the interhemispheric effect causes an amplification of the effects of
obliquity cycle on the Earth's surface. In other words, the real Earth has a larger response
to the obliquity cycle in the ocean dominated Southern Hemisphere, as compared to the
hypothetical symmetric Earth. Consequently, the interhemispheric effect of continental
geography contributes to the muting of precessional signal and amplification of obliquity
signal recorded in paleoclimate proxies such as benthic $\delta^{18}$O isotope records.
There are various ways in which the Earth's continental asymmetry affects climate. Here,
we have shown how these interhemispheric effects influence the Earth's climate response
to orbital forcing via the radiative and atmospheric dynamical processes represented in a



slab-ocean GCM. While computationally challenging, future work should include
complimentary simulations with AOGCMs, to explore the potential modifying role of
ocean dynamics on the amplifying and weakening interhemispheric responses to orbital
forcing demonstrated here.



**Table 1. Experimental Setup of Model Boundary Conditions and Forcings**

| Run ID | LSX Configuration | Eccentricity | Obliquity | Precession [a] | GHGs |
|---|---|---|---|---|---|
| CONTROL$_{NHSP}$ | Modern | 0.034 | 23.2735 | 270° (NHSP) | Preindustrial |
| CONTROL$_{SHSP}$ | Modern | 0.034 | 23.2735 | 90° (SHSP) | Preindustrial |
| CONTROL$_{HIGH}$ | Modern | 0.034 | 24.5044 (HIGH) | 180° | Preindustrial |
| CONTROL$_{LOW}$ | Modern | 0.034 | 22.0425 (LOW) | 180° | Preindustrial |
| NORTH-SYMM$_{NHSP}$ | North-symmetric | 0.034 | 23.2735 | 270° (NHSP) | Preindustrial |
| NORTH-SYMM$_{SHSP}$ | North-symmetric | 0.034 | 23.2735 | 90° (SHSP) | Preindustrial |
| NORTH-SYMM$_{HIGH}$ | North-symmetric | 0.034 | 24.5044 (HIGH) | 180° | Preindustrial |
| NORTH-SYMM$_{LOW}$ | North-symmetric | 0.034 | 22.0425 (LOW) | 180° | Preindustrial |
| SOUTH-SYMM$_{NHSP}$ | South-symmetric | 0.034 | 23.2735 | 270° (NHSP) | Preindustrial |
| SOUTH-SYMM$_{SHSP}$ | South-symmetric | 0.034 | 23.2735 | 90° (SHSP) | Preindustrial |
| SOUTH-SYMM$_{HIGH}$ | South-symmetric | 0.034 | 24.5044 (HIGH) | 180° | Preindustrial |
| SOUTH-SYMM$_{LOW}$ | South-symmetric | 0.034 | 22.0425 (LOW) | 180° | Preindustrial |

**NHSP:** Northern Hemisphere Summer Solstice at Perihelion
**SHSP:** Southern Hemisphere Summer Solstice at Perihelion
[a] Orbital precession in the GCM is defined here as the prograde angle from perihelion to
the Northern Hemispheric vernal equinox.




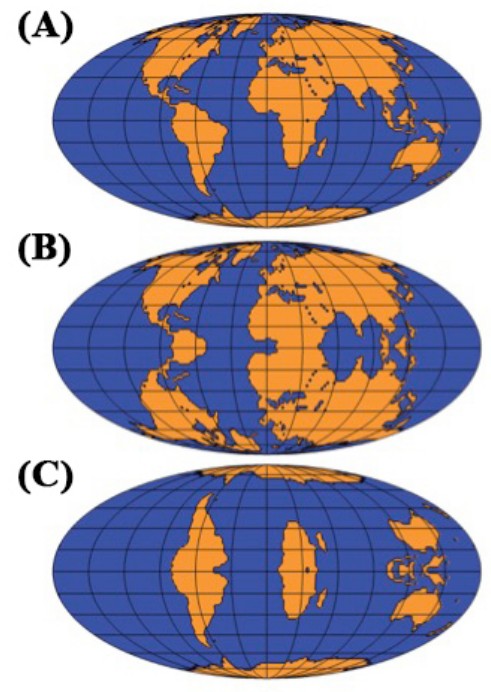


**Figure 1.** (A) Modern continental geography (B) NORTH-SYMM geography and (C)

SOUTH-SYMM geography



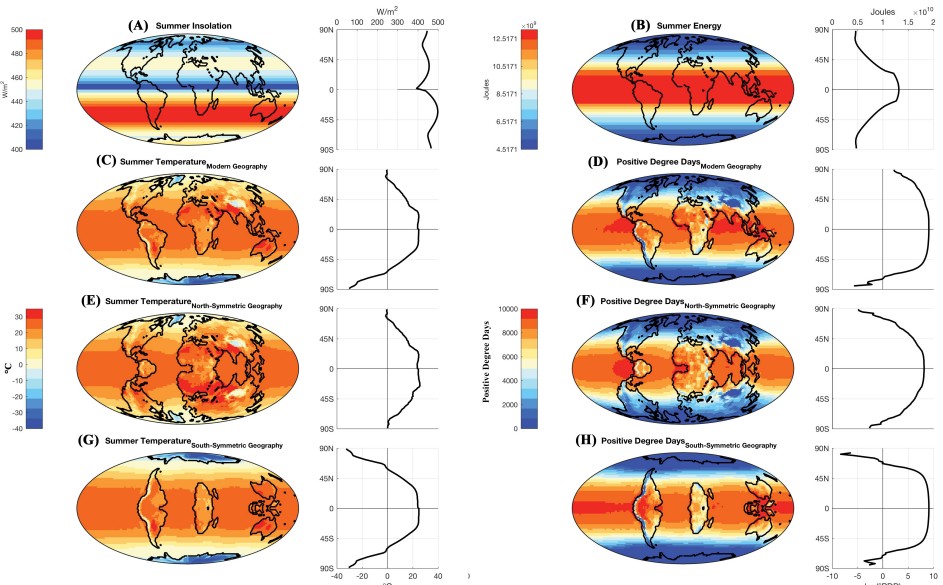

**Figure 2.** (A-D) Demonstration of Earth's asymmetric climate response to symmetric climate forcing. Simulations are forced by modern day orbit: (A) Summer insolation; (B) summer energy; (C) Summer Temperature; and (D) PDD. (E-H) Demonstration of Earth's symmetric climate response to climate forcing when idealized symmetric Earth geographies are used. Simulations are forced by modern day orbit: (E) and (F) Summer Temperature and PDD for NORTH-SYMM simulation, (G) and (H) Summer Temperature and PDD for SOUTH-SYMM simulation. The zonal averages are plotted on the right of each Figure. Zonal averages of PDD are plotted on a log scale.



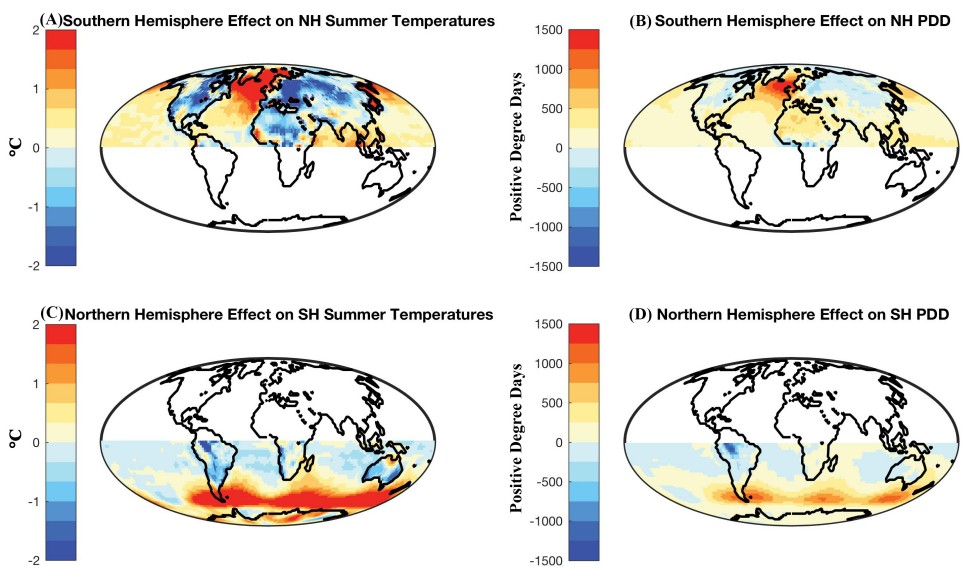

**Figure 3.** Interhemispheric effect of Southern Hemisphere continental geography on (A) Northern Hemisphere Summer Temperature (ST) and (B) Positive Degree Days (PDD). Interhemispheric effect of Northern Hemisphere continental geography on (C) Southern Hemisphere Summer Temperature (ST) and (D) Positive Degree Days (PDD). Zonal averages are plotted on the right of each figure.



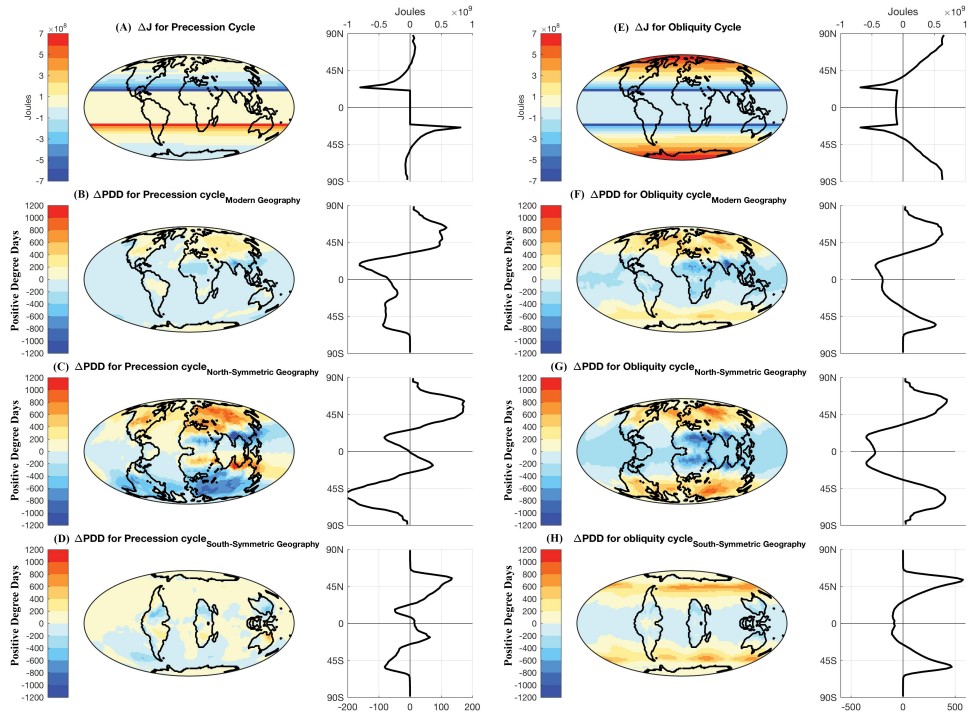

**Figure 4.** (A) Summer Energy change for a transition from SHSP to NHSP orbit and the corresponding change in Positive Degree Days in CONTROL (B); NORTH-SYMM (C) and SOUTH-SYMM (D) simulations. (E) Summer Energy change for a transition from LOW to HIGH orbit and the corresponding change in PDD in CONTROL (F); NORTH-SYMM (G) and SOUTH-SYMM (H) simulations.



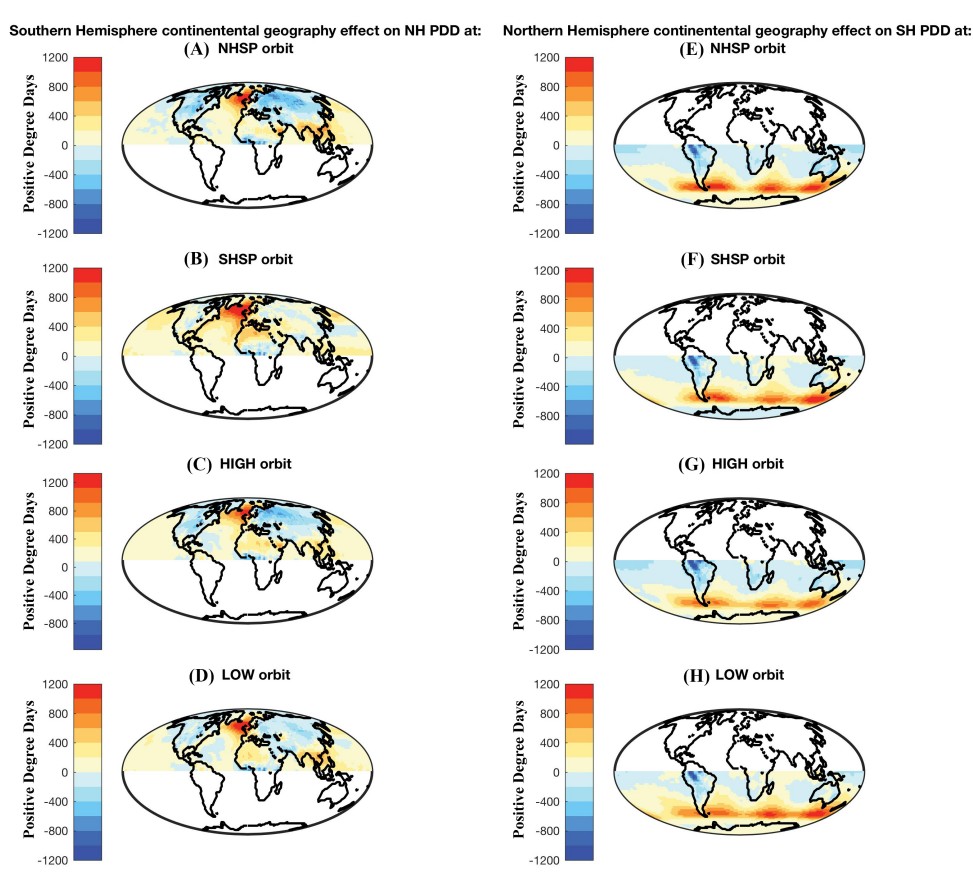

456

**Figure 5.** Interhemispheric effect of Southern Hemisphere continental geography on Northern Hemisphere climate: (A) at NHSP $[(\widehat{e_{PDD}})_{NHSP}]$; (B) at SHSP $[(\widehat{e_{PDD}})_{SHSP}]$; (C) at HIGH $[(\widehat{e_{PDD}})_{HIGH}]$; (D) at LOW $[(\widehat{e_{PDD}})_{LOW}]$.

Interhemispheric effect of Northern Hemisphere continental geography on Southern Hemisphere climate: (E) at NHSP $[(\widehat{e_{PDD}})_{NHSP}]$; (F) at SHSP $[(\widehat{e_{PDD}})_{SHSP}]$; (G) at HIGH $[(\widehat{e_{PDD}})_{HIGH}]$; (H) at LOW $[(\widehat{e_{PDD}})_{LOW}]$.



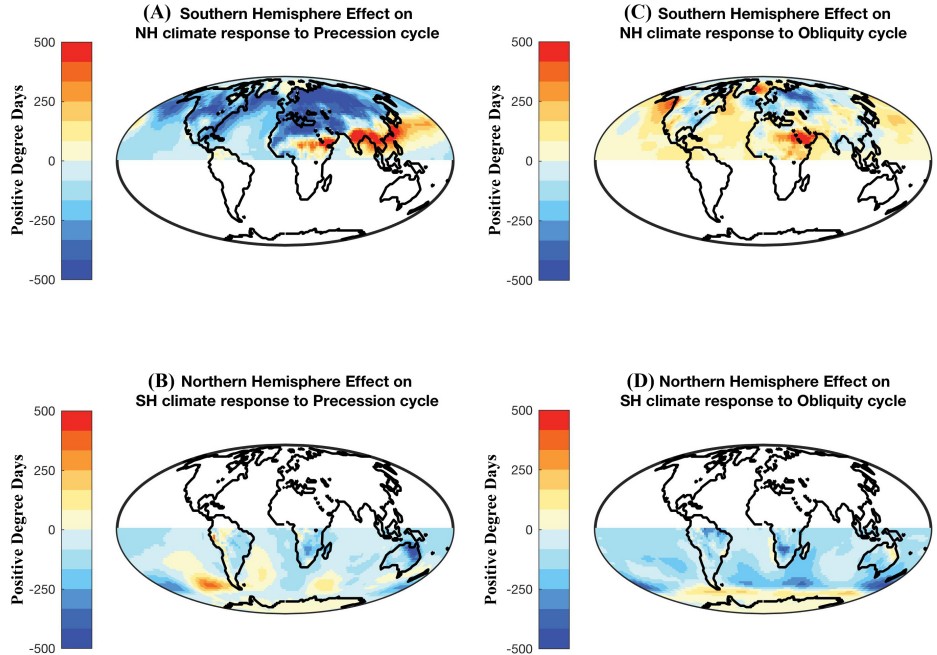

463

**Figure 6.** Interhemispheric effect of: (A) Southern Hemisphere continental geography on

Northern Hemisphere $\Delta$PDD$_{\text{precession}}$ (response to precession forcing) $[(\widehat{e_{PDD}})_{precession}]$,

(B) Northern Hemisphere continental geography on Southern Hemisphere $\Delta$PDD$_{\text{precession}}$

(response to precession forcing) $[(\widehat{e_{PDD}})_{precession}]$, (C) Southern Hemisphere continental

geography effect on Northern Hemisphere $\Delta$PDD$_{\text{obliquity}}$ (response to Obliquity)

$[(\widehat{e_{PDD}})_{obliquity}]$, (D) Northern Hemisphere continental geography effect on Southern

Hemisphere $\Delta$PDD$_{\text{obliquity}}$ (response to Obliquity) $[(\widehat{e_{PDD}})_{obliquity}]$.



Alder, J. R., Hostetler, S. W., Pollard, D. and Schmittner, A.: Evaluation of a present-day climate
simulation with a new coupled atmosphere-ocean model GENMOM, Geosci. Model Dev., 4(1),
69–83, doi:10.5194/gmd-4-69-2011, 2011.
Barron, E. J., Thompson, S. L. and Hay, W. W.: Continental distribution as a forcing factor for
global-scale temperature, Nature, 310(5978), 574–575, doi:10.1038/310574a0, 1984.
Berger, A. and Loutre, M. F.: Insolation values for the climate of the last 10 million years, Quat.
Sci. Rev., 10(4), 297–317, doi:10.1016/0277-3791(91)90033-Q, 1991.
Croll, J.: No Title, Philos. Mag. J. Sci., 39(259), 81–106, 1870.
Crowley, T. J. and North, G. R.: Paleoclimatology, Oxford University Press. [online] Available
from: http://books.google.com/books/about/Paleoclimatology.html?id=VDE-
mKySpM0C&pgis=1 (Accessed 13 November 2014), 1996.
DeConto, R.: Plate Tectonics and Climate Change, in Encyclopedia of Paleoclimatology and
Ancient Environments SE  - 188, edited by V. Gornitz, pp. 784–798, Springer Netherlands., 2009.
Deconto, R. M., Pollard, D., Wilson, P. A., Pälike, H., Lear, C. H. and Pagani, M.: Thresholds for
Cenozoic bipolar glaciation., Nature, 455(7213), 652–6, doi:10.1038/nature07337, 2008.
Fawcett, P. J. and Barron, E. J.: The Role of Geography and Atmospheric CO2 in Long Term
Climate Change: Results from Model Simulations for the Late Permian to the Present, in Tectonic
Boundary Conditions for Climate Reconstructions, pp. 227–247, Oxford University Press., 1998.
Flato, G. M. and Boer, G. J.: Warming asymmetry in climate change simulations, Geophys. Res.
Lett., 28(1), 195–198, doi:10.1029/2000GL012121, 2001.
Hay, W. W.: Tectonics and climate, Geol. Rundschau, 85(3), 409–437, doi:10.1007/BF02369000,

492     1996.

Hay, W. W., Barron, E. J. and Thompson, S. L.: Results of global atmospheric circulation
experiments on an Earth with a meridional pole-to- pole continent, J. Geol. Soc. London., 147(2),



385–392, doi:10.1144/gsjgs.147.2.0385, 1990.
Huybers, P.: Early Pleistocene glacial cycles and the integrated summer insolation forcing.,
Science, 313(5786), 508–11, doi:10.1126/science.1125249, 2006.
Kang, S. M., Held, I. M., Frierson, D. M. W. and Zhao, M.: The Response of the ITCZ to
Extratropical Thermal Forcing: Idealized Slab-Ocean Experiments with a GCM, J. Clim., 21(14),
3521–3532, doi:10.1175/2007JCLI2146.1, 2008.
Kang, S. M., Seager, R., Frierson, D. M. W. and Liu, X.: Croll revisited: Why is the northern
hemisphere warmer than the southern hemisphere?, Clim. Dyn., 1457–1472, doi:10.1007/s00382-
014-2147-z, 2014.
Kiehl, J. T., Hack, J. J., Bonan, G. B., Boville, B. a., Williamson, D. L. and Rasch, P. J.: The
National Center for Atmospheric Research Community Climate Model: CCM3*, J. Clim., 11(6),
1131–1149, doi:10.1175/1520-0442(1998)011<1131:TNCFAR>2.0.CO;2, 1998.
Koenig, S. J., DeConto, R. M. and Pollard, D.: Pliocene Model Intercomparison Project
Experiment 1: implementation strategy and mid-Pliocene global climatology using GENESIS
v3.0 GCM, Geosci. Model Dev., 5(1), 73–85, doi:10.5194/gmd-5-73-2012, 2012.
Loutre, M.-F., Paillard, D., Vimeux, F. and Cortijo, E.: Does mean annual insolation have the
potential to change the climate?, Earth Planet. Sci. Lett., 221(1–4), 1–14, doi:10.1016/S0012-
821X(04)00108-6, 2004.
Loutre, M. F.: Clues from MIS 11 to predict the future climate – a modelling point of view, Earth
Planet. Sci. Lett., 212(1–2), 213–224, doi:10.1016/S0012-821X(03)00235-8, 2003.
Philander, S. G. H., Gu, D., Lambert, G., Li, T., Halpern, D., Lau, N.-C. and Pacanowski, R. C.:
Why the ITCZ Is Mostly North of the Equator, J. Clim., 9(12), 2958–2972, doi:10.1175/1520-
0442(1996)009<2958:WTIIMN>2.0.CO;2, 1996.
Raymo, M. E., Lisiecki, L. E. and Nisancioglu, K. H.: Plio-Pleistocene Ice Volume, Antarctic



Climate, and the Global d18O Record, , 313(July), 492–495, 2006.
Short, D. A., Mengel, J. G., Crowley, T. J., Hyde, W. T. and North, G. R.: Filtering of
Milankovitch Cycles by Earth's Geography, Quat. Res., 35(2), 157–173, doi:10.1016/0033-
5894(91)90064-C, 1991.
Stone, P. H.: Constraints on dynamical transports of energy on a spherical planet, Dyn. Atmos.
Ocean., 2(2), 123–139, doi:10.1016/0377-0265(78)90006-4, 1978.
Stouffer, R. J., Manabe, S. and Bryan, K.: Interhemispheric asymmetry in climate response to a
gradual increase of atmospheric CO2, Nature, 342(6250), 660–662, doi:10.1038/342660a0, 1989.
Thompson, S. L. and Pollard, D.: Greenland and Antarctic Mass Balances for Present and
Doubled Atmospheric CO 2 from the GENESIS Version-2 Global Climate Model, J. Clim.,
10(5), 871–900, doi:10.1175/1520-0442(1997)010<0871:GAAMBF>2.0.CO;2, 1997.
Trenberth, K. E., Fasullo, J. T. and Kiehl, J.: Earth's Global Energy Budget, Bull. Am. Meteorol.
Soc., 90(3), 311–323, doi:10.1175/2008BAMS2634.1, 2009.