# Peer review of "Interhemispheric Effect of Global Geography on Earth's Climate Response to"

_Climate of the Past, 2017_

## Referee Comment (RC1) · Anonymous Referee #1 · 29 Jun 2017

The authors examine the role of hemispheric asymmetry of land masses on the climate response to orbital forcing using a slab ocean model. The main weakness of the paper is lacking of physical understanding. The authors merely describe the model response without any physical interpretation. It makes me to wonder what is the goal of this study. If there is no physical understanding behind, the results are presumably model dependent. In a similar context, the results are likely to be modified when the ocean model is employed, as the authors acknowledge. Then, to be a meaningful work, it is critical to physically understand the findings before adding complexity to a model.

Other minor comments are following: 1. Why are the authors focus on the Positive

[Figure]

Degree Days (PDD)? In the abstract, the authors need to specifically mention that the impacts on the PDD are addressed among many other climate variables.

2. line 43-48: The asymmetrical land masses can cause hemispheric differences in many features such as albedo, strength of water vapor feedback, cloud distribution, and ocean circulation. Which feature is most important?

3. I think the introduction is separated from the main text. I suggest adding the goal of the present study, which is something like lines 157-167 in Section 3, and making connections to the following results.

4. Line 93: Need to specify the mixed layer depth.

5. Is the land model interactive? Or is vegetation prescribed?

6. Line 137: Omit "defined as"

7. Line 226-230: In comprehensive models, the polar amplification only appears in the Arctic and the warming signal is less distinctive in the Antarctica than in the Arctic. Would this asymmetrical land masses be a part of the cause? The authors may be able to add some discussions about it.

8. Figure 3, and alike: Why is one hemisphere masked out? There could be two way interaction, so I wonder what the response is like in the other hemisphere. Rather than masking out, I suggest showing the global response.

9. Line 204-205: The warming effect in the North Atlantic will weaken the Atlantic Meridional Overturning Circulation. Add some discussions about it.

---

## Referee Comment (RC2) · Anonymous Referee #2 · 11 Jul 2017

In this manuscript, the authors are investigating the role of geographic hemispheric symmetry or asymmetry on Earth climate. This question was raised already in the XIXth century and could indeed be an interesting scientific discussion. Unfortunately, as it stands, 1 - the paper does not provide a sufficient scientific background on this problem, 2 – does not present the severe limitations of the methodology used, 3 – does not discuss the mechanisms involved in their results, 4 – does not provide sufficient discussion for the theoretical interpretation of their work. As it stands, the resulting manuscript is merely a presentation of some AGCM simulations, without much scientific content.

[Figure]

1 - Overall, it is not clear what the main focus of the paper is.

1a: If it is a theoretical paper, then:

The authors should discuss in much greater details their critical assumptions, and how they impact the results. In particular, there is no dynamical ocean in their model set-up. A key theoretical finding in the 1980s was that the ocean heat transport cannot be symmetrical even under symmetric boundary conditions (see eg. Bryan, 1986; Thual et al, 1992, . . .) : this symmetry breaking leads to multiple equilibria in the Atlantic deep circulation, and is now quite well understood on a theoretical point of view. So it is unlikely that a true coupled climate model would react symmetrically under symmetric boundary conditions, as found here with an atmospheric component only. This single omission is probably enough to reject the paper as it is now.

For a theoretical paper, it is not sufficient to get "almost symmetrical results" (line 174; line 267) without further comments. Indeed, it would be rather easy to go a step further and to give a proper account of the "remaining asymmetry". The obvious sources of asymmetry are linked to the astronomical forcing, in an explicit way through the precession and the location of the perihelion (as discussed rapidly in the paper), but also in an implicit way through the choice of a calendar. This last point is quite critical but is not even mentioned in the paper. For instance, (line 194) the authors do not specify if the number of summer days is fixed according to the present conventional calendar or if it is astronomically defined. Since this last specific point is at the center of the discussion on climate symmetry/asymmetry since the XIXth century (see below), it is a pity that the authors fail to address or even mention this key question.

The choice of an "averaged eccentricity" is not a very good one for a theoretical discussion on symmetry/asymmetry, since the role of precession will not be very clear, as discussed above (results are "almost symmetric"). Using a high eccentricity would provide an easier discussion of the asymmetry linked to the astronomical forcing. Using a zero eccentricity would remove the precessional forcing and, by difference, allow for a

true discussion of the role of asymmetry in the forcing.

1b: If the main topic is about paleoclimate changes, then:

The authors should explain how these symmetric/asymmetric configurations relate to actual past changes of the Earth geography. There is no real mention about this, expect in the introduction: "the global continental configuration has been close to its present form since the mid-Cenozoic Âż (line 29) and in the conclusion Âń the amplification (or weakening) of the response to insolation changes at precessional and obliquity periods might explain some of the important features of late Pliocene-early Pleistocene climate variability Âż (line 408). These two statements contradict each other : how the insignificant geographic changes of the late Cenozoïc could modify the response of the Earth system to astronomy ? Are these (insignificant) changes towards a stronger or a weaker symmetry ? Are we talking about amplification or weakening ? No only the manuscript is unclear on this conclusion, but it seems also quite contradictory. My impression is that the main topic of the paper is not about actual past climate changes.

1c: If the main topic is about atmospheric mechanisms involved in the simulated symmetrical/asymmetrical response of the atmosphere, then the authors should discuss these mechanisms. As observed also by Reviewer #1, this is obviously not the case here.

So my overall impression is that the paper could possibly be a rather theoretical one on symmetry/asymmetry of climate. Unfortunately, the key ingredients are missing.

2 - The title of the paper is about Earth hemispheric asymmetry, in the context of astronomical forcing. I was expecting some introduction to explain why this question is so central in the astronomical theory of paleoclimate. Unfortunately, there is no such background in the manuscript. This was a heavily debated topic in the XIXth century, with many important astronomical, physical, theoretical discussions (see eg. Lyell, 1830). The key point at that time was to understand whether the Southern hemisphere was colder because of the land-sea distribution (asymmetry) or because of the astronomical forcing. This ended up with various ideas on present-day climate and various astronomical theories of Quaternary climates. So, indeed, the question of symmetry/asymmetry was quite fundamental since the beginning, and indeed it has not been investigated thoroughly with GCM tools. But as it stands, the paper does not provide a good introduction and a good theoretical incentive to look at the simulations results.

3 – The ocean is represented by a slab ocean. This is quite a critical assumption that should be discussed (see above). But some details should also be provided here: is there any representation of an oceanic meridional heat transport ? How is it parameterized ? Does such a parameterization reproduces present day SSTs ? Is it (almost) symmetrical when the atmospheric results are (almost) symmetrical ?

4 – line 174-175: "with some small remaining asymmetry due to the current timing of perihelion". Indeed, this is precisely what is called the astronomical forcing linked to precession, as understood since the XIXth century, and to the "present day" calendar convention.

5 – line 187 and line 214: "the effect of Southern Hemisphere continental geography on Northern Hemisphere climate" (and reciprocally). Obviously, the authors are mixing up the notions of "effect" (something which involves causality, mechanisms, etc. . .) and the notion of "correlation" based simply on plotting maps without even discussing if there might be a causal link between the Southern continental geography and the Northern climate. This sentence is very symptomatic of the whole paper. Measuring such an "effect" (as attempted in this manuscript) has no real scientific value if there is nothing behind it, but mere coincidence.

6 – The choice an "extreme precession" (line 260) makes the implicit assumption that only high latitudes are involved in the question of symmetry/asymmetry. This is quite strange, since monsoons are likely strongly involved in this problem, through interhemispheric heat transport changes, something that could be captured by an atmospheric only model. But then, a more natural choice for "extreme precession" might

correspond to perihelion at equinoxes (not at solstices). Precession is fundamentally a cycle, and a circle has no "extremes" unless you look it from a specific point of view.

7 – line 299: "According to Milankovitch theory, the Northern Hemisphere should experience 'interglacial' conditions when perihelion coincides with boreal summer Âż. line 400: "At precessional periods, at which the high latitude summer intensity primarily varies Âż. Obviously, the authors have misunderstood Milankovitch. According to Milankovitch (1941), the "summer" intensity (defined as the caloric summer insolation) depends primarily on obliquity (therefore his prediction of 41 ka glacial cycles), not on precession.

8 – line 255: citation of Raymo 2006 for precession being out of phase in both hemisphere. Maybe Herschel (1835) or Lyell (1830) would be more appropriate, though scientists were aware of that fact certainly much earlier, probably since Hipparchus ($\sim$ 30 BC). Citing a recent paper for a very old (simple geometric) relationship seems quite inappropriate to me.

9 – line 121 et 168: meriodionally -> meridionally

References:

Lyell, C. Principles of Geology, vol.1, 2, 3. 1st ed. J. Murray, London (1830) pp. 511. Bryan, F. High-latitude salinity effects and interhemispheric thermohaline circulations. Nature (1986) vol. 323 pp. 301-304 Thual, O. and McWilliams, J. The catastrophic structure of thermohaline convection in a two-dimensional fluid model and a comparison with low-order box model. Geophys. Astrophys. Fluid Dyn. (1992) vol. 64 pp. 67-95 Milankovitch. Kanon der Erdbestrahlung und seine Andwendung auf das Eiszeitenproblem. (1941) pp. 633 Herschel, J. On the Astronomical Causes which may influence Geological Phænomena. Transactions of the Geological Society of London (1835) (2) pp. 293-299

---

## Author Response (AR1)

**To,**

**Editor – Climates of the Past**

At the outset, we would like to thank the editor and the reviewers for their comments. While we appreciate the general sentiment of the reviewer's comments, this initial paper was never intended to provide a detailed dynamical analysis of the Land Asymmetry Effect. Instead, it is the first paper to point out that such a LAE exists. To our knowledge, this is the first modeling study exploring the response of symmetrical Earth geometries to orbital forcing. The results are tantalizing and we hope they spark future analysis and study, which includes our own, currently underway.

Because a full dynamical analysis was never the intention of this first manuscript on the topic, the GCM simulations used relatively course resolution and only saved select meteorological fields as time averaged (monthly means). The output is not suitable to exploring some atmospheric dynamical processes that would be interesting to explore in the context of the LAE patterns. With that said, we have added a few select examples of meteorological fields that correlate strongly with the LAE patterns of response. These include clouds, snow cover, and sea ice, which provide a strong regional (radiative) amplifying or dampening effect of local orbital forcing. The far field influence of the opposite hemisphere's geometry has not been determined, but this would be a tall order even for a modern climatological study, as such inter hemispheric teleconnection patterns remain poorly understood.

We hope the editor views this manuscript as simply a first step, reporting on interesting GCM results that use a unique geographic set up to illustrate that there are far field influences of global geography that moderate/accentuate the Earth's response to orbital forcing. The manuscript was never intended to report on anything more than just that.
Thanking you,
Rajarshi Roychowdhury, Rob DeConto

**Response to Anonymous Referee #1**

We fully acknowledge the limitations of using a slab-ocean GCM to study the effect of hemispheric asymmetry on the climate response to orbital forcing. These limitations are addressed directly in the text. We also stress that the slab-ocean configuration also has important advantages: its computational efficiency allows a range of orbital conditions to be explored, while simplifying interpretations of the results and minimizing ocean-model dependencies. We hope this initial work sparks further studies using fully coupled models to further quantify the results shown here. We stress that giving a fully mechanistic explanation of the hemispheric effects is beyond the scope of this initial short paper. This is the focus of ongoing work and a forthcoming manuscript. However, we have extended the discussion in our revised paper based on other observable parameters from the model simulations. In the revised paper we show that the observed hemispheric effects on climate are related to asymmetry in clouds, snow cover, and surface pressure patterns that impacts heat transport (e.g. Figures 1 and 2).

1. We chose PDD as one of our climate variables because both temperature and the duration of summer are important for Earth's climate response. In this case, Positive Degree-Days are calculated as PDD $= \sum_i \propto_i T_i$, *where $T_i$ is the mean daily temperature on day i, and α is one when $T_i \geq 0°C$ and zero otherwise.* The PDD captures the extremity as well as the duration of the warm season.

As requested, we will include the statement that we address other climate variables along with PDD in our paper in the abstract.

**2.** After doing additional analyses, we conclude that cloud fraction, liquid water content in the atmosphere and pressure are the most important. We extend the discussion towards causes of the observed hemispheric effect in the revised paper.

3. We agree with the Reviewer's suggestion, and we have rewritten our introduction accordingly.

4. The mixed layer depth is 50m, and will be specified in the manuscript.

5. The land model is not interactive, and the vegetation is prescribed. This was done purposefully to simplify interpretations of the results.

6. We have corrected the same.

7. The muted polar amplification in Antarctica observed in models may be caused in part by the asymmetrical landmasses between Northern and Southern Hemispheres. However, the Land asymmetry effect has a dependence on the specified orbit (astronomical configuration), which in turn might alter the effect on polar amplification.

8. The reason we chose to mask out one hemisphere in the figures is because the hemispheric effect is calculated differently for Northern and Southern Hemispheres, i.e. for the effect in Northern Hemisphere; the Southern Hemisphere is made symmetric, and vice-versa. We can update the figures as per Reviewer's suggestion and show the global response in single maps for both Northern and Southern Hemispheres.

9. Based on our model simulations, we observe that there is a positive warming effect in the North-Atlantic Ocean, and in general the Northern Hemisphere oceans are slightly warmer relative to a symmetric Earth. However, as mentioned in the paper, our model does not capture the explicit changes in ocean currents and the deep ocean. Hence we refrain from making any additional comments on the behavior of the AMOC or any other global ocean current, which would likely alter greatly in a symmetric or near-symmetric Earth.

[Figure]

Figure 1: The hemispheric effects on 3D Cloud Fraction.

[Figure]

Figure 2: The hemispheric effects on fractional snow cover

**Response to Anonymous Referee #2**

1 – As clearly acknowledged in the body of the manuscript, we agree with Referee#2 that there are limitations with not using a fully coupled atmosphere-ocean GCM. We hope to spark future studies using true coupled models to study the interesting role of geographical hemispheric asymmetry on Earth's climate. While there are limitations to our model, its computational efficiency has the advantage of allowing a wide range of orbital parameter space to be explored, while minimizing ocean-model dependencies on the results.

In line 174 and 267, the "almost asymmetrical results" refer to almost symmetric results for positive degree days (PDD). This asymmetry arises from precession and location of perihelion with respect to each hemisphere's summer, as mentioned in the paper and also pointed out by the Reviewer. However, the calculation of PDD is autonomous of the choice of calendar. The definition of PDD, or the analogous Summer Energy (total integrated summer insolation, as defined in Huybers 2006) depend on a fixed threshold of daily average temperature to determine the duration and timing of summer. The choice of calendar does become important when we consider average summer temperatures. In our present results, we assumed a modern calendar, and defined summer in Northern Hemisphere as June-July-August and in Southern Hemisphere as December-January-February. Following the reviewers recommendation we have modified the approach by choosing a summer definition based on an insolation threshold (See point 4).

For the simulations shown here, the choice of averaged eccentricity was empirical, rather than theoretical. As rightly pointed out by Anonymous Referee #2, using zero eccentricity would make the orbit circular, thus muting any effect of precession. This is undesirable, as we wish to include the possible effects of astronomical forcing in the measured hemispheric asymmetry effect. On the other hand, using a high eccentricity intensifies the effect of precession, which may enhance its influence on the measured asymmetry. Please note that we are not discussing the role of hemispheric asymmetry on the TOA insolation forcing itself (because it is same regardless of the continental arrangement on Earth). Instead, we wish to discuss the role of hemispheric asymmetry on the climate, at different insolation forcings (corresponding to different orbits). Thus, a 'true discussion' of the role of asymmetry would involve simulations at every possible orbital configuration, including all possible eccentricity values. Keeping in mind the concise format of this paper, we show our results with a representative value of averaged eccentricity (0.034).

However, we would like to mention that our conclusions regarding the hemispheric effects are not modified by using a different value of eccentricity (the values of individual model grid cells vary in the final figures, but the spatial patterns remain the same).

1b. Our paper does not focus on any specific time period in the past. We regret the confusion in our wording, and would like to clear any contradictory statements we might have inadvertently made. In line 408, we mention: "the amplification (or weakening) of the response to insolation changes at precessional and obliquity periods might explain some of the important features of late Pliocene-early Pleistocene climate variability". We do not mean that any geographical change due to plate tectonics has led to modification of the Earth's response to astronomical forcing. What we intend to stress is that the asymmetric continental configuration has an important control on the climate response of the Earth that might be relevant to interpretations of Plio-Pleistocene climate variability based on proxy records. Here we refer to specific climatic features of the Plio-Pleistocene, such as the dominance of obliquity over precession in the 40-kyr world benthic isotope records. The glacial cycles during the late Pliocene to early Pleistocene (~1-3 myr) had dominant 40-kyr frequencies. The primary frequency associated with the benthic $\delta 18O$ records from this period corresponds to variation in the obliquity phase. This raises a major contradiction to Milankovitch's theory of orbital forcing, which predicts precession should be the strongest frequency in glacial-interglacial cycles. Raymo (2006) suggested that the glacial cycles are controlled by local summer insolation (dominated by the 23-ky precession period), but are out-of-phase between Northern and Southern Hemispheres. In addition to this, we suggest that in each hemisphere, the precessional effect on ice-volumes is muted due to hemispheric asymmetry (Roychowdhury and DeConto, Nature Communications, 2017, in review). When summers are warm in one hemisphere due to precession (precession varied in isolation, obliquity kept constant), the hemispheric asymmetry makes it colder than expected, and when it is cool due to precession, the interhemispheric asymmetry makes it warmer. We regret the confusion caused due to our vague wording, and have rephrased our statements in our revised manuscript to remove any such confusion.

2 – We thank the Referee for his valuable suggestion to include a more comprehensive introduction. We have rewritten the introduction and provide a stronger theoretical incentive to investigate the land symmetry/asymmetry problem using a GCM framework.

3. This caveat is fully acknowledged in the manuscript. In this case, this well tested and often used slab ocean model calculates prognostic (fully varying) SSTs as a function of seasonal thermodynamics. Ocean heat transport is parameterized as a function of the local sea surface temperature gradient, the fraction of land and sea at a given latitude, and tuned to fit the modern latitudinal dependence of ocean heat convergence with respect to latitude. Because the ocean depth is limited to 50-m (enough to capture the seasonal cycle of the mixed layer), the GCM comes into equilibrium relatively quickly, allowing us to run many experiments under a wide range of orbits. While a study like this would ideally include a full depth dynamical ocean, we view this as a next step, hopefully motivated in part by the results published here. Furthermore, dynamical ocean models introduce an additional level of complexity and complex model-dependencies that we think are best avoided in this initial study.

4. The choice of calendar affects the calculation of summer temperatures in our simulations with varying precession. In today's orbital configuration, the Earth is at perihelion during Southern Hemisphere summer (SHSP). This coincides with Northern Hemisphere summer occurring when the Earth is at aphelion. During NHSP, the earth is at perihelion during Northern Hemisphere summer. Consequently, in the latter case, the duration of NH summer season is shorter than present. This is due to Kepler's laws, which states that the time elapsed between the two positions of the Earth along the ellipse are proportional to the area covered. Thus, due to precessional effects amplified by eccentricity changes, the length of seasons varies through time (Joussaume and Braconnot, 1997, etc). When summer occurs at perihelion, the duration of summer is short, but the intensity of TOA insolation is strong. When summer occurs at aphelion, the duration of summer is long but the intensity is weaker. To take into account the duration of summer, Peter Huybers suggested the use of a time integrated summer metric (Huybers, 2006). In our manuscript, we have used PDD (following the definition from Huybers 2006 paper) as a measure of climate response, and this metric is independent of the choice of calendar.

However, when we discuss the hemispheric effects in "summer temperatures", we need to address the question of defining a calendar for different orbits. To better account for the phasing of the insolation curves for different orbits, instead of seasons defined with the same length as modern, we now define seasons by an insolation threshold; which will account for the astronomical positions as well as the phasing of the seasonal cycle of insolation. In this case, we define summer as the period during which the average daily insolation is above a specified threshold (325 $W/m^2$). [Figure 3 and 4]

5. This paper indeed focuses on measuring the 'effect', with an assumption of causal link between the Southern Hemisphere geography and Northern climate and vice-versa. Giving a comprehensive mechanism of the hemispheric effect is beyond the scope of this particular manuscript. However, we have investigated the main linkages between the hemisphere effect and various atmospheric processes. As noted in the revised paper, we find that clouds, fractional snow cover, liquid water content in the atmosphere and atmospheric heat transport has the strongest impact of hemispheric asymmetry, thus contributing to the net hemispheric land asymmetry effect.

6. This is an excellent point raised by Reviewer#2. Our choice for "extreme precessions" being the solstices stems from our original motivation for studying hemispheric asymmetry at the poles. In the revised manuscript, we add new simulation results with perihelion coinciding with the solstices. This is a substantial improvement.

7. Line 299: "According to Milankovitch theory, the Northern Hemisphere should experience 'interglacial' conditions when perihelion coincides with boreal summer"
We regret the confusion caused here by the lack of clarity in our wording. What we meant is that when precession is considered in isolation, i.e. not considering the compounding effect of obliquity, then perihelion coinciding with Northern summer would imply warm 'interglacial' type conditions in Northern Hemisphere. This wording has been changed.

Line 400: "At precessional periods, at which the high latitude summer intensity primarily varies."
We implied summer insolation intensity, and not the caloric summer insolation (which is an integrated measure of insolation over time). The summer insolation intensity varies at precessional periods (23kyr) (Raymo et al. 2006, Huybers 2006, etc.). The caloric summer half-year at 65N, defined as the energy received during the half of the year with the greatest insolation intensity also has more than half its variance in the precession bands (Milankovitch 1941, Huybers and Tziperman 2008, etc.). This has been clarified in the revised manuscript.

8 – We agree with Referee #2's observation, and have updated the manuscript with historical references wherever applicable in the manuscript

9 – We will correct this in a revised manuscript.

*References*

Joussaume, S. and Braconnot, P, 1997; Sensitivity of paleoclimate simulation results to season definitions

Huybers, P. 2006; Early Pleistocene Glacial Cycles and the Integrated Summer Insolation Forcing

Huybers, P. and Tziperman, 2008; E. Integrated summer insolation forcing and 40,000-year glacial cycles: The perspective from an ice-sheet/energy-balance model

Raymo, M. E., Lisiecki, L. E. and Nisancioglu, K. H. 2006; Plio-Pleistocene Ice Volume, Antarctic Climate, and the Global d18O Record

[Figure]

Figure 3: Insolation curves for different orbits for Northern and Southern Hemispheres. The horizontal line shows the threshold of 325 W/m2 used to define summer. And day with average insolation higher than 325 W/m2 is considered as summer. Defining summer this way accounts for the variation in duration and timing of summer at different orbits.

[Figure]

Figure 4: Hemispheric Effects on Summer Temperature (summer defined by an insolation threshold) for different orbits

(a) Northern Hemisphere summer coinciding with Perihelion (b) Southern Hemisphere summer coinciding with Perihelion (c) High Obliquity orbit (d) Low obliquity orbit.

In each panel,

Top: Interhemispheric effect of Southern Hemisphere continental geography on Northern Hemisphere Summer Temperature (ST), summer defined as above.

Bottom: Interhemispheric effect of Northern Hemisphere continental geography on (C) Southern Hemisphere Summer Temperature (ST), summer defined as above.

---

## Author Response (AR2)

**To,**

**Editor – Climates of the Past**

We would like to thank the editor and the reviewers for their comments. Our primary intent with this paper, as described in the introduction, is to report on the differing response of symmetric versus asymmetric geographies to orbital forcing. We acknowledge that this first step is indeed mainly descriptive and does not provide a detailed dynamical analysis of the Land Asymmetry Effect. We reiterate that this study (saving monthly mean GCM output) was never intended to be a detailed dynamical examination of the climatic response to changing geographies under different orbits and that's way beyond the scope of the paper. However, we do observe interesting patters related to clouds, snow cover, cross-equator sea-level pressure (wind) pattern that impacts sea ice, etc. which then amplifies the response. However, these interesting patterns require additional study, and we hope some well-designed dynamical experiments could be conducted to tease out physical mechanisms behind the correlations in the future.

We hope the editor views this manuscript as simply a first step, reporting on interesting GCM results that use a unique geographic set up to illustrate that there are far field influences of global geography that moderate/accentuate the Earth's response to orbital forcing.

Thanking you,
Rajarshi Roychowdhury, Rob DeConto

**Response to Review #1:**

The authors made significant efforts to attempt to respond to the requests of the reviewers. They must be praised for that. Nevertheless this does not seem fully satisfactory to me since some of the important requests are rejected as 'irrelevant' (which I can partially understand) or worse, the additional parts add more confusion. The section about the clouds, the liquid water content, atmospheric pressure, snow cover and sea ice should inform the reader about the feedbacks but this is not really the case. Firstly, the analysis is almost exclusively limited to correlations and, on the other hand, it lacks an explanation of what the variables really are (is it annual average, seasonal, other?) and what is the physics behind the correlation.

The GCM simulations used for this study are relatively course resolution and only saved select meteorological fields as time averaged (monthly means). The output is not suitable to exploring some atmospheric dynamical processes that would be interesting to explore in the context of the LAE patterns, limiting us with few select examples of meteorological fields that relate with the observed LAE. These include clouds, snow cover, and sea ice, which provide a strong regional (radiative) amplifying or dampening effect of local orbital forcing. We have extended our analysis based on these variables, and have added the explanation of the variables used in the figures.

The authors wrote themselves that a 'true discussion' of the role of asymmetry would involve simulations at every possible orbital configuration, including all possible eccentricity values'. And it really puzzles me because they have only 6 experiments, all with the same eccentricity. We are really far from covering the whole field of orbital parameters (and I understand that it is impossible for a GCM). However, what is done here is probably statistically not significant. Moreover, 'statistical methods' (latin hypercube) are very efficient to identify the set of variables that would be more representative for the full ranges of combinations although, even a carefully designed set of experiments would mean too many experiments.

The aim of this first paper was to point out that the Land Asymmetry Effect exists, and we extend the results to show that the LAE is different at various orbital configurations. As mentioned above, it is impossible to cover the entire orbital range using a GCM. Choosing a set of orbital parameters is also impractical, as any one combination of orbital parameters will not be representative of the orbital range. For phenomena that depend strongly on the correlation of two or more variables, Latin hypercube sampling has little advantage over traditional Monte Carlo sampling. And being limited by computational resources, we cannot run an extremely high number of orbital combinations. Thus, we chose a different approach to run the simulations at highest and lowest obliquity, and 4 precessional configurations. We choose the mean value of eccentricity, which is the most frequent value of eccentricity in the last 2.0 million years. The value of eccentricity (0.034) is chosen from the plot of the kernel density estimate of the eccentricity values from the last 2.0 million years.

[Figure]

Figure: KDE Distribution plot of the Eccentricity values from the last 2.0 million years. The selected value of eccentricity (0.034) is shown by the black vertical line.

**Response to Review #2:**

*1 – Quite generally, this paper still falls short of my expectations in terms of result analysis. It is quite frustrating to see interesting model results without any explanation of the physical processes involved. Compared to the previous version, the authors are now presenting correlations with some key variables. This might be a first step, but the manuscript still remains quite descriptive on this part.*

The GCM simulations used for this study are relatively course resolution and only saved select meteorological fields as time averaged (monthly means). The output is not suitable to exploring some atmospheric dynamical processes that would be interesting to explore in the context of the LAE patterns, limiting us with few select examples of meteorological fields that relate with the observed LAE. These include clouds, snow cover, and sea ice, which provide a strong regional (radiative) amplifying or dampening effect of local orbital forcing. We have extended our analysis based on these variables, along with the physical processes involved.

*2 – Slab ocean. I do very well understand that a slab ocean is computationaly more efficient : My comments did not concern the use of a slab ocean in this study, but the discussion of its impact on the results. Is the parameterized ocean heat flux symmetric when temperatures and continents are symmetric? If so, then it should be explained clearly in the text. If not, this might be a serious problem.*

Yes, the parameterized ocean heat flux is symmetric in our symmetrical Earth simulations. Poleward oceanic heat flux is defined as a function of the temperature gradient and the zonal fraction of land and sea at given latitude in the model. In the model, the ocean heat flux parameterization is based only on observed estimates. This allows the model to be used for paleoclimatic applications with very different ocean configurations than the present. (Thompson and Pollard, 1997). A description of ocean heat transport symmetry is now included in the manuscript.

[revised manuscript text omitted]

---

## Author Response (AR3)

**To,**

**Editor – Climates of the Past**

We would like to thank the editor and the reviewers for their comments and deciding to proceed with this manuscript. We hereby submit the updated manuscript with the desired revisions. The output data from this study has been uploaded and data availability has been added to the manuscript.

Thanking you,

Rajarshi Roychowdhury, Rob DeConto

**Response to Review #1:**

Fig1: All the curves are annual mean. Is it correct? This should be specified

Fig 1(a) is annual mean and has been corrected in the manuscript. Fig 1(b) is for summer only (mentioned), and Fig 1(c) is sum of Positive Degree Days (defined in the main body of the manuscript).

GCM: Does this stand for Global Climate Model or General Circulation Model? What is the difference between both?
We use GCM for Global Climate Model. We have corrected inconsistencies in the manuscript.

L 133: What does geography mean in this context? A reference may be useful.
Added relevant details to the manuscript.

L 150: Summer insolation is obviously NOT symmetric across both Hemisphere according to Figure 3a. It remains still unclear what is the difference between insolation and energy in this context.
We have edited the text for clarity, and added the definition for summer energy along with reference.

L 227: Why is it 'Northern and austral summers' and not 'boreal and Southern summers'? I would have expected 'Northern and Southern summers' or 'boreal and austral summers'. What is the difference?
We have corrected the inconsistencies in the manuscript.

L 239: 'The forcing (summer energy (J)) calculated at the top of the atmosphere is symmetric across both hemispheres (Figure 5a)'. According to the figure, it seems rather anti-symmetric or asymmetric.
The forcing (Summer Energy) is numerically symmetric across both hemispheres, but out-of-phase, as expected for a change in precession (Since precession impacts both hemisphere in opposite ways). We have edited the text for clarity.

L 258: The section numbering should be checked throughout.
We have corrected the inconsistencies in the manuscript.